# Hybrid Approach for Detecting Propagandistic Community and Core Node on Social Networks

**Akib Mohi Ud Din Khanday** [1], **Mudasir Ahmad Wani** [2,*], **Syed Tanzeel Rabani** [1] and **Qamar Rayees Khan** [1]

1   Department of Computer Sciences, BGSB University, Kashmir 185234, India
2   EIAS Data Science Lab, Prince Sultan University, Riyadh 11586, Saudi Arabia
*   Correspondence: mwani@psu.edu.sa

**Abstract:** People share their views and daily life experiences on social networks and form a network structure. The information shared on social networks can be unreliable, and detecting such kinds of information may reduce mass panic. Propaganda is a kind of biased or unreliable information that can mislead or intend to promote a political cause. The disseminators involved in spreading such information create a sophisticated network structure. Detecting such communities can lead to a safe and reliable network for the users. In this paper, a Boundary-based Community Detection Approach (BCDA) has been proposed to identify the core nodes in a propagandistic community that detects propagandistic communities from social networks with the help of interior and boundary nodes. The approach consists of two phases, one is to detect the community, and the other is to detect the core member. The approach mines nodes from the boundary as well as from the interior of the community structure. The leader Ranker algorithm is used for mining candidate nodes within the boundary, and the Constraint coefficient is used for mining nodes within the boundary. A novel dataset is generated from Twitter. About six propagandistic communities are detected. The core members of the propagandistic community are a combination of a few nodes. The experiments are conducted on a newly collected Twitter dataset consisting of 16 attributes. From the experimental results, it is clear that the proposed model outperformed other related approaches, including Greedy Approach, Improved Community-based 316 Robust Influence Maximization (ICRIM), Community Based Influence Maximization Approach (CBIMA), etc. It was also observed from the experiments that most of the propagandistic information is being shared during trending events around the globe, for example, at times of the COVID-19 pandemic.

**Keywords:** dubious; propagandistic; community; core; nodes; online social networks

## 1. Introduction

Unlike the traditional web, online social networks, driven mainly by content, treat users as first-class citizens. A user joins a network, posts content, creates friendly ties and remains connected with other network users in the network [1]. This fundamental user-to-user link structure supports online engagement by offering a framework for organizing both real-world and virtual contacts, identifying content and expertise that has been provided or recommended by friends, and discovering other users with similar interests [2]. Blogs, content aggregation sites, internet fora, online social networks, and phone data records provide real-time data [3–5]. New technologies are required to obtain, handle, and evaluate these data.

In recent years, social community research has depended extensively on online interaction data and explicit linking in online social community platforms such as Facebook, Twitter, LinkedIn, Flickr, and Instant Messenger [6]. Twitter is one of the most popular social media platforms, with millions of active users from almost every country and generating a revenue of USD 3.7 billion (https://www.businessofapps.com/data/twitter-statistics/, accessed on 21 November 2020). Twitter is a real-time, highly social micro-blogging site

that allows users to publish short status updates known as tweets, which are displayed on timelines. In their 280-character content, tweets may contain one or more entities and references to one or more places in the world. Twitter is a public platform with millions of tweets published daily from millions of user accounts, all of which have educational and commercial value. The Twitter API allows developers to access Twitter's data [7]. Understanding users, tweets, and timelines is crucial in successfully using Twitter's Application Program Interface (API). In Twitter, there are three types of APIs: Search API, Streaming API, and Representational State Transfer (REST) API. In this research work, a sample Twitter propaganda network was created and utilized to investigate several community detection algorithms. The list of friends and followers of propaganda is the most important piece of information that was extracted. Propaganda is a kind of biased or unreliable information that can mislead or intend to promote a political cause. The disseminators involved in spreading such information establish accounts, particularly fake identities [8,9], and create a sophisticated network structure.

Socio-metrics measure social relationships to investigate network structure. The structure of the network is determined by checking the quality of interconnections, role of entities, information flows, network evolution, clusters/communities in a network, nodes in a cluster, the cluster/center network's node, and nodes on the periphery [10]. Based on interaction modules, characteristic values, and the expectation of undetected connections among nodes, the functioning of related items from network groups is detected [11]. In communities, the nodes have various relationships with one another [12]. Identifying a community is a difficult task that involves grouping nodes into small communities, and a node in a community structure may belong to several communities at the same time [13].

Some of the main contributions of this work are as follows:

- An AI-based framework is proposed for detecting propagandistic communities and propagandists.
- Leader Ranker Algorithm is used for mining the candidate nodes within the interior of a community.
- Constraint Coefficient is used for mining candidate nodes within the boundary of a community.
- The Boundary-based Community Detection Approach (BCDA) has been proposed to identify the core nodes in a propagandistic community based on the interior and boundary nodes, which over-performed the existing approaches such as Degree Discount, Greedy, etc., in terms of running time.
- A novel dataset collected from the Twitter blogging site was generated out of this study. This dataset consists of 16 attributes considered favorable for propagandistic community detection research. The dataset will be available for the researcher of this domain.

The article is divided into six sections. Section 1 gives a brief overview of propaganda, Twitter, and the need for finding propagandistic communities on social networks. Sections 2 and 3 discuss the relevant work and background knowledge about propaganda and community structure on social networks. The proposed methodology is discussed in Section 4; Section 5 shows the experimental results. In Section 6, the work is concluded and also few research directions are provided.

## 2. Related Work

Humans have the nature to make communities in the real world and the same is reflected in social media. A study [14] proposed an algorithm for detecting group structure. The two major characteristics of the algorithm are (1) remove edges from the network iteratively, forming communities from the networks; (2) recalculate the edges after each removal. The algorithms used are more effective at detecting group structure in both machine and real-world network data. There is a need for improving the algorithm as the utilized algorithm becomes intractable for larger systems. The algorithm has been improved to reduce computational complexity. Ref. [15] proposed an algorithm for detecting a

network's group structure. The algorithm has two key functions: the first is to remove edges from the network iteratively so that communities can be created from the networks, and the second is to recalculate the edges after each removal. For both real and real-world network results, the algorithm performed efficiently. In [16], the researchers found the hidden networks of political influence within the government and tracked the evolution of political groups. Network construction, community discovery, and community evolution tracking were the three key processes they used to accomplish this. The findings of the study could be used to create a political culture observation method that aids public oversight of political power transitions for better checks and balances in democratic societies. The authors in [17] developed a novel method for detecting Stealthy accounts in online social networks. Node-level community identification, features, classification, and finding stealthy sybils are some of the steps in their approach. The research can be expanded upon to find new approaches to deal with malicious accounts based on OSN users' community-based features. Researchers in [18] proposed a new algorithm for detecting groups in social networks in order to obtain useful and relevant data. According to the findings, the Modularity-based approach outperformed the Eigenvector-based approach. Complex and dense networks with overlapping nodes and cross-edges were not included by the authors.

The study in [19] proposed a new method that employs ontology and clustering algorithms. They perform relation analysis and group detection in each cluster. There are five modules in this method: 1: Preparation of a social network dataset. 2: Text preprocessing and data modeling. 3: Clustering of social objects. 4: Partitioning of social network members. 5: Examining the connections. They obtained a dataset consisting of 517,431 emails from 151 Enron Company employees. As dimensions, they receive 506 keywords. They eliminated terms with similar meanings after using their definition. There are 414 words left after this. There was a higher level of success with their system. Other clustering algorithms that work on the basis of weight can be used. The proposed approach is also applicable to dynamic graph datasets.

Researchers proposed a new algorithm [20] for detecting communities using Graph mining techniques. They begin by creating a Community incidence matrix. The occurrence matrix was then used to determine the number of communities. The group graph can be detected using the community number series. Isolated groups may also be identified. They tested their algorithm by detecting groups in different villages. This method for detecting communities is easy and efficient. With the support of tags, likes, and retweets, this technique can be used on social media sites. Another similar study [21] proposed the Attention Automaton, a probabilistic finite automaton that can estimate a user community's collective attention. The Communities on Twitter are focused on users' geographical proximity or shared interests (such as followers of a specific account). They discovered that the likelihood is determined by two factors: 1. The inclination of the user group to change their focus. 2. The categorical affinity of the user community. They came up with the term "volatility" to describe the inclination for people's attention to change based on time slots. In addition, different user groups respond to patterns in different categories in different ways. They used GT-TTL (Graphical Location—Trending Topic List) and BT-TTL (Brand Audience—Trending Topic List) to conduct various experiments. They choose 30 locations around the world at random in GT-TTL. The Attention Automaton performed 44 percent better than ARIMA and 71 percent better than random selection in terms of F-score results. In BT-TTL, 30 consumer groups of brands were chosen at random. Overall, the Attention Automaton outperformed the random scheme by 38 percent and the ARIMA model by 74 percent in terms of F-score. They used patterns as an attention function, but it would be fascinating to see what other social network assets might be used in the same way. It is an intriguing challenge for patterns that go beyond hashtags, such as multi-word expressions such as "America Loves Justin Bieber", which falls into the category (Location + Entertainment). In terms of game theory, it will be interesting to learn how patterns fight to break into the $TjjTL$. Authors in [22] detected groups based on shared interests, user engagement, and social events. This method necessitates seed user and

friend list user data. Various parameters about the seed consumer are extracted, such as the number of tags and general interests. They proposed a formula for calculating the value of tag-like and tag statements. After determining the typical social activity value and number of tags, the data is fed into the k-clustering algorithm. Communities are represented by the clusters formed. The dataset of 121 users with their social activities was used as the input data. The equations are applied to the dataset as part of the method. The CSA (Common social activity) parameter is developed. The CSA and number of tags are then fed into the K-clustering algorithm. The societies are created by clusters. A semantic-based approach can be used to more accurately determine the user's area of interest. The method can be used to identify propagandistic communities. Furthermore, there are several studies carried out to detect other groups on social networks, for example, the study in [23] has proposed a way to identify suspicious groups on the Facebook network. Two similar studies [24,25] have been proposed to predict fake user communities and regions respectively based on emotions. The present study is mainly focused on the detection of propagandistic communities and core nodes on social networks. The following Conclusion has been drawn from the above literature reviews.

- The work on finding propagandistic communities is in its infancy. To our knowledge, very less literature has been found regarding the same.
- Traditional algorithms are used for detecting community structure.
- Less work has been conducted on Social Media data.
- The core members responsible for sharing the propaganda among communities need to be identified to help law enforcement agencies break the chain.

## 3. Background Knowledge

There are two types of community detection techniques: agglomerative and divisive. A graph with no edges is gradually filled in with them using Agglomerative approaches. A stronger edge is used as a source to lend strength to a weaker one. In divisive approaches, a full graph has its edges eliminated successively [26–28]. There can be any number of communities of varied sizes within a given network. These qualities make it exceedingly challenging to detect communities [29–31]. However, other strategies have been presented in the field of community detection. Some popular approaches that are used for detecting communities in a network are Newman–Girvan and Random Walk. The below section discusses these approaches in detail.

### 3.1. Newman and Girvan Algorithm

Newman and Girvan Algorithm is a general approach for locating communities. It divides the vertices without requiring the number of communities to be specified. The algorithm has three distinct features: Firstly, Edges are gradually eliminated from a network. Secondly, edges to be deleted are determined by computing betweenness scores and lastly, betweenness ratings are recalculated once each edge is removed. This algorithm comes under divisive methods in which the number of shortest paths going through the edge determines the edge weight. The degree of one edge's influence over other vertices in a network is a generalization of the central vertex betweenness measure. The number of edge statements vertex as their terminus equals the number of shortest paths via the vertex [32]. In order to implement the Girvan–Newman algorithm, the following steps need to be followed:

1. Compute edge betweenness for every edge in the graph.
2. Take away the edge with the greatest edge betweenness.
3. Compute edge betweenness for remaining edges.
4. Repeat steps 2–4 until all edges are removed.

The key idea behind the Girvan–Newman method is "edge betweenness", or the total number of shortest paths that traverse an edge in a network. The algorithm starts with one source vertex $sinV$ in a graph, calculates edge weights for pathways that pass through it, and then repeats the process for each vertex in the graph and adds the weights for each

edge. Implementing this algorithm to a tree, a form of a graph, with only one shortest path from the source vertex to any other vertex. Starting from the leaf nodes, the edges that connect them to the rest of the vertices in a tree are given the value one because there is only one shortest path to node(s) travelling through that edge [33]. The edge weight value determines the number of shortest paths in the tree from the source vertex to every other vertex going through a specific edge. The edge betweenness for each edge is obtained by repeating the process for each vertex and computing the sum of weight values for each edge. The algorithm implements two methods for calculating edge betweenness. (1) The distance from the source vertex is assigned to each vertex by using a breadth-first search to identify the number of shortest paths from the source to the vertices. An abstract data type queue is used to implement this algorithm component efficiently. (2) Begins with edge incident to the vertex as the start point and ends with maximum distance covered from the source vertex as the endpoint. The number of shortest paths travelling through it is determined for each edge. The $(d_i, w_i, b_i)$ is calculated mathematically for each vertex $i$ in $V$, where $d_i$ is the distance from the source vertex, $w_i$ is the number of shortest paths from the source vertex to vertex $i$, and $b_i$ is the number of shortest paths between the source vertex and any vertex in the graph that passes through vertex $i$. Assume that $Adj(v)$ is the set of all vertices adjacent to $v$ such that $v \in V$. The second phase of edge betweenness calculation begins from the last vertex noted in the first phase and works backwards through the vertices visited in the first phase. From the source, only one shortest path passes through the last marked vertex [34].

For all source vertices $s$, both phases of the algorithm are run, and edge betweenness for each edge is determined as the sum of the edge betweenness calculated in each step. This component of the method has a computational complexity of $O(en)$, where $e$ is the number of edges and $n$ is the number of vertices. The edge with the highest edge betweenness is deleted after each edge betweenness calculation, and the method is repeated until there are no more edges. As a result, the Girvan and Newman algorithm has a complexity of $O(2en)$ [35].

### 3.2. Random Walk

A random walk in the network can be used to capture the network structure and detect communities in a big complicated network. It is based on the idea that a network will tend to stay trapped in a denser region or community for longer. This concept is utilized to bring nodes into the community. In graph theory, random walk is the process of randomly visiting a neighboring node from the source node and continuing the process throughout the network. The random walk process is analogous to the Markov chain algorithm in which the collection of states corresponds to the visited path's vertices [36].

Let $G = (V, E)$ be a directed graph and $v_0$ be the random walk's beginning node mathematically. The random walk is located at node $i$ at the $t_{th}$ step and moves from node $i$ to node $j$ at the $(t + 1)$th step, with a transition probability of $1/d_i$, where $d_i$ is the degree of node $i$. The transition matrix $T(k)$ represents the chance of reaching all nodes from all other nodes in the network through a k-length random walk. The odds of visiting all other nodes from node $i$ in the $k$ walk length corresponds to each tuple in the transition matrix [37]. These probabilities are based on the network's structural information. The following conclusions can be taken from the network's structure:

- If two nodes $i$ and $j$ belong to the same community, the likelihood of accessing node $j$ from $i$ is greater than visiting a node outside the community. Even though the likelihood is high, this does not imply that they are members of the same community.
- Because the walker tends to visit vertices with high degrees, the probability $T(k)_{i,j}$ is dependent on the degree of $j$.
- Two vertices in the same community have a tendency to see all other vertices in the same way, and $T(k)_{i,e} \approx T(k)_{j,e}$, $\forall i, j \in$ same community and $e \in [1, n]$.

A transition matrix is derived from a random walk through the graph for detecting community. The probability of visiting each node from every other node in $k$ steps is

described by the transition matrix. The probability of visiting node $j$ from $i$ in $k$ steps is represented by $T(k)_{i,j}$. The random walk transition matrices $T(1)$, $T(2)$, $T(3)$, and $T(k)$ correspond to 1, 2, 3, and $k$ walk lengths, respectively. The following Equation (1) defines the probability of transitioning from vertex $i$ to vertex $j$ in a one-length random walk:

$$T_{i,j}^1 = A_{i,j}/d_i \tag{1}$$

where $A_{i,j}$ is the adjacency matrix of the network and $d_i$ is the degree of vertex $i$.

When contrasted to nodes outside the communities, a node belonging to the same community would behave similarly. Any two nodes within a community have the same appearance as the rest of the nodes in the network. Based on the walk length $k$, the transition matrix $T(k)_{i,j}$ is used to determine the similarity between two vertices. The likelihood of reaching one node from another would vary for different travel lengths. The Euclidean distance between row vectors corresponding to nodes $i$ and $j$ in a matrix can be used to compute the similarity between $i$ and $j$ for $k$ walk length. The similarity between $i$ and $j$ for $k$ walk length can be computed by the Euclidean distance between row vectors corresponding to nodes $i$ and $j$, in matrix $T(k)$. Equation (2) defines the same: [38]

$$S(i,j) = \sqrt{\sum_{l=i}^{n}(T(k)_{i,l} - T(k)_{j,l})^2 / dl} \tag{2}$$

The main aim of this community detection method is to calculate the similarity of a node based on a random walk in the network. In the worst scenario, the temporal complexity of this algorithm is $O(2en)$, where $e$ is the number of edges and $n$ is the number of nodes in the network [39].

## 4. Materials and Methods

Online social networks, particularly large-scale social networks such as Twitter, can exhibit a distinct community structure with highly coupled nodes. However, it is essential to mention that nodes within different communities mostly share a weak connection. The computation complexity may significantly decrease by finding seed/core nodes within communities [40,41]. The method consists of three phases:

- Propaganda Detection;
- Community Detection;
- Influence Maximization (IM) of community structure.

Figure 1 shows the general framework for detecting Core member and community structure from propaganda tweets.

**Propaganda Detection:** Propaganda is detected using various Machine Learning Classifiers. Data are extracted from Twitter using its Application Program Interface (API), and an annotation scheme proposed by [42] is used for annotating the tweets into Propaganda and Non-Propaganda classes. Preprocessing is performed using techniques such as Tokenization, Stemming, Lemmatization, stopword removal, etc. [43]. A hybrid feature selection technique is used for selecting relevant features. TF/IDF, Bag of Words, Tweet Length, and Sentimental feature selection techniques were merged in this work. Support Vector Machine, Decision Tree, and Multinomial Naive Bayes classifiers were used for classification. The result showed that the Support Vector Machine showed better accuracy than other machine learning classifiers.

**Influence maximization:** Identifying the k-size subsets of node $SinV$ is the objective of IM. The diffusion model maximizes the influence spread f(S), for the given network "$G = (V, E)$" and the size of seed nodes $k$ [44–47].

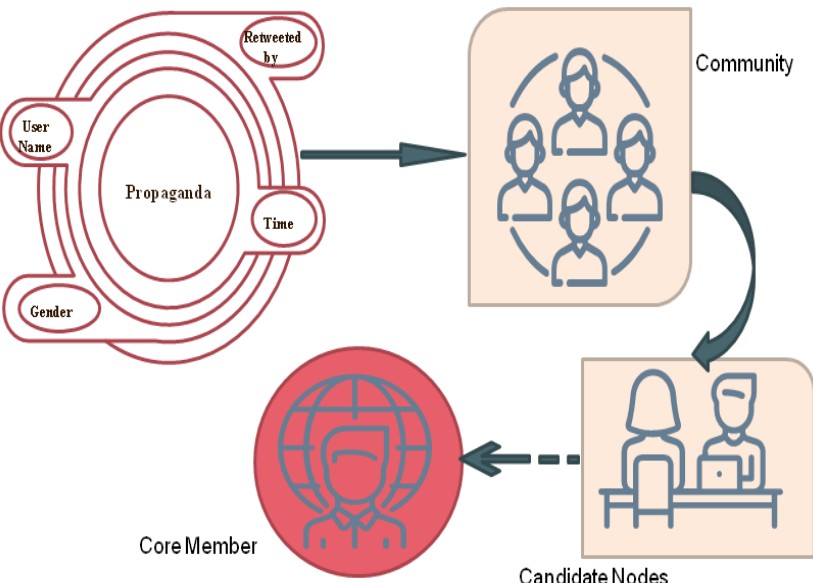

**Figure 1.** Community and Core Member Detection Framework.

**Community Structure:** A cluster of nodes having higher edge densities within them and lower edge densities between groups is represented by a group in a network with $G = (V, E)$ by its community structure. The proportion of edges is often known as modularity $Q$. It is a measure which frequently assesses the strength of the community structure, which is calculated using Equation (3):

$$Q = \frac{1}{2|E|} \sigma(C_u, C_v) \sum_{u,v \in V} \left( e_{uv} - \frac{d_u d_v}{2|E|} \right) \tag{3}$$

where $|E|$ represents number of edges.

$\sigma(C_u, C_v)$ signifies that the function has a value of 1 if nodes u and v are in the same community, else it has a value of 0.

$d_u$ is the degree of node $u$.

$e_{uv}$ represents the direct edge between node $u$ and $v$.

$Q$ is modularity, and in real social networks, its value ranges from 0.3 to 0.7. Larger modularity score, stronger community structure.

**Community Detection:** Using the traditional k-means approach on our dataset, the network is partitioned into different communities, as depicted in Figure 2. The algorithm is based on the steps listed below:

1. $k$ clustering centers are chosen at random.
2. Calculate how similar each point is to the center.
3. Organize the points into clusters if their similarity is below the threshold.
4. Repeat steps 2 and 3 until the center is unmodified, then update the cluster centers.

The modularity and number of communities are calculated using [48] algorithm on our dataset. The results showed that modularity equals 0.490, and about six Communities were present. The Graphical representation of modularity with several nodes is given in Figure 3. The x-axis of Figure 3 determines the modularity score in decimals, and the y-axis shows the number of nodes present in the respective communities (c1, c2, ..., c6).

**Candidate Node Set:** Candidate node sets are the highly dominating nodes in a community. They may be present either within the community or on the boundary. Figure 4 represents one of the communities (C6) detected from the dataset, on the same community, the proposed approach is used to identify the core nodes of the community.

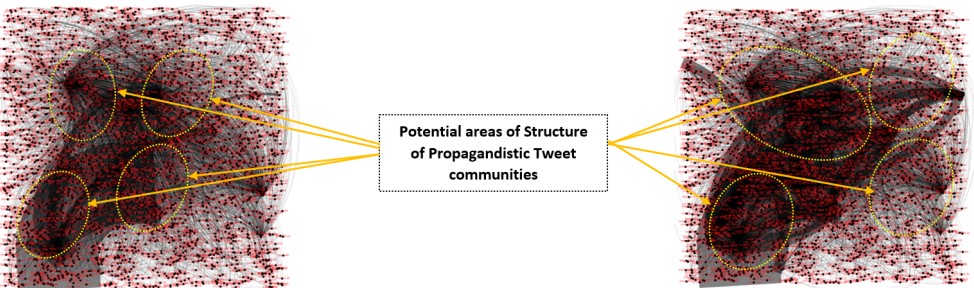

**Figure 2.** Network Structure of Propagandistic Tweets.

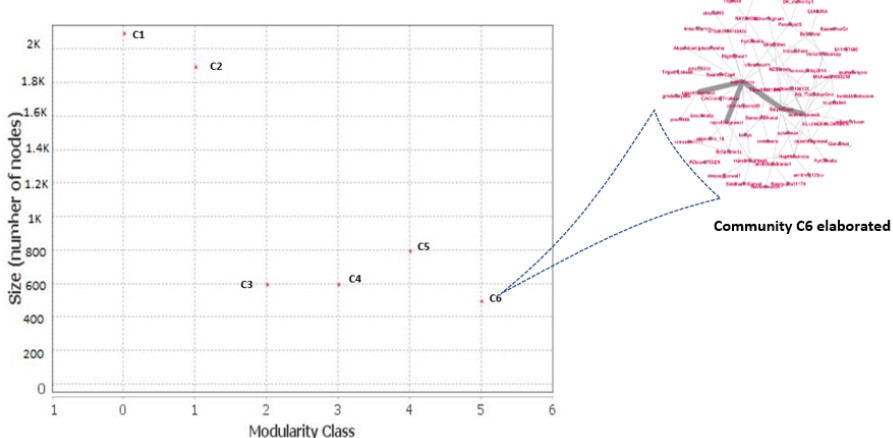

**Figure 3.** Number of Communities detected in the Dataset with their modularity scores.

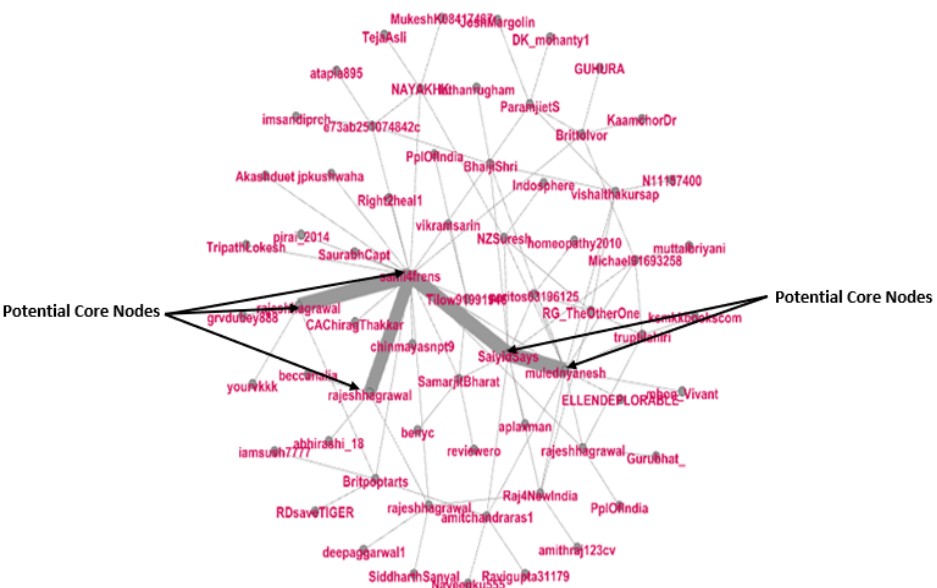

**Figure 4.** Community (C6) with Labels and core nodes detected from the Dataset.

## 4.1. Finding Candidate Nodes within the Community

For finding candidates from the interior of the community, we used the Leader Ranker algorithm, which improves the Page Rank algorithm. The said algorithm improves the convergence and measures the influence of nodes within the community. All nodes in the network, with the exception of node $g$, are first given a unit of LR (Leader Rank Value), which is then distributed evenly to the nodes' neighbors. It can be conceptualized mathematically as a random walk on a directed network using the stochastic matrix P,

where the probability of a random walk from $i$ to $j$ is given by $p_{ij} = \frac{e_{ij}}{k_i}$. Then, based on the following Equation (4), all nodes' LRs are updated.

$$LR_i(t+1) = \sum_{j \in N_i} \frac{LR_j(t)}{k_j} \tag{4}$$

where $N_i$ represents the adjacent node set of node $i$.

$k_j$ represents the degree of node $j$.

$LR_j(t)$ represents the LR of node $j$ at iteration $t$.

Until it converges, it works in an iterative manner. Equation (5) defines the final LR value of node $i$ after $LR$ of the ground node has been averagely distributed among other nodes.

$$LR_i = LR_i(t_c) + \frac{LR_g(t_c)}{N_i} \tag{5}$$

where $LR_i(t_c)$ is the LR value of node $i$ at iteration $t_c$.

$LR_g(t_c)$ represents the LR value of $g$ at iteration $t_c$, $N_i$ represents adjacent nodes of node $i$. $t_c$ represents the convergence point. According to the above analysis, nodes having higher LR values are more significant. The nodes with higher significance are chosen within the community to generate candidate nodes.

### 4.2. Finding Candidate Nodes from the Boundary

The structural holes are the voids between individuals that lack direct connections, as illustrated in Figure 5. The location of hole C in the structure serves as a bridge between two nodes. It has been recognized that the link between communities is tenuous, and those structural nodes with the benefit of weak ties are the key to the transmission of influence between communities. To create candidate nodes, thus, structural holes are identified from the boundaries of each community.

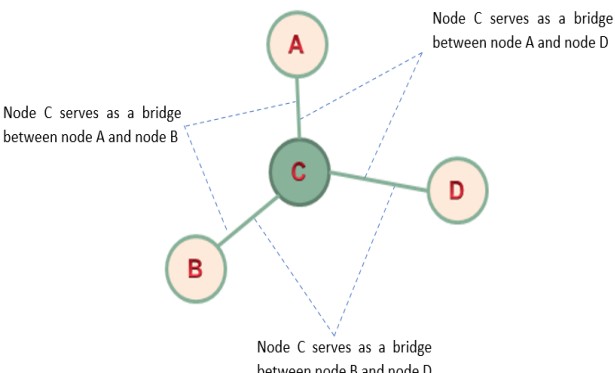

**Figure 5.** Structural Node from Boundary of a Community.

Network Constraint Coefficient (CT) measures the constraints imposed by forming the structural holes. CT is calculated using the Equation (6)

$$CT_i = \sum_{j \in N_i} \left( p_{ij} + \sum_q p_{iq} p_{qj} \right)^2 \tag{6}$$

where node $q$ is the common neighboring node of $i$ and $j$, $p_{ij}$ is the cost associated in a direct way, and $p_{iq}$ is the cost associated in an indirect way. In the unweighted network, $p_{ij}$ is usually equal to $\frac{e_{ij}}{\sum_{j \in N_i} e_{ij}}$.

Hub nodes that have a large number of neighboring nodes are more likely structural holes. By ignoring the characteristics of large social network communities, the network constraint coefficient just considers the network topology. Nodes present in many communities may act as a bridging element, allowing information to propagate to the majority of

the communities. As a result, the network CT and community structure are specified as indicated in Equation (7).

$$OC_v = \frac{10^{-CT_v} \odot Nb(v)}{maxOC} \tag{7}$$

where $CT_v$ is the constraint coefficient set of node v, $Nb(v)$ is neighbor set of node v, $maxOC$ is the normalization factor. The *OC* is utilized for evaluating the influence potential of the boundary nodes. Then, the candidate node set is comprised of the border nodes with greater influence.

*4.3. Core Nodes Selection*

After choosing candidate nodes within the interior as well as the boundary of a community. Core nodes are selected using the sub-modular property-based Greedy approach. It is estimated using Equation (8).

$$f_c(S) = f(C \cap S, G_C) \tag{8}$$

where "$f(C \cap S, G_C)$" denotes the number of activated nodes by the set of core nodes "$C \cap S$" in sub-network $G_C$. As the candidate nodes' influence is limited to the communities in which they and their neighbors reside.

The marginal impact increment of interior nodes is determined using Formula (9).

$$f_C(u|S) = f(u \cap (C_u \cap S), G_{C_u}) - f(C_u \cap S, G_{C_u}) \tag{9}$$

$$s.t.C_u = \sum_{C \in C_S, C \cap u \neq \phi} C$$

The marginal influence increment of boundary nodes is determined using the following Formula (10).

$$f_C(u|S) = f(u \cap (C_u \cap S), G_{C_u}) - f(C_u \cap S, G_{C_u}) \tag{10}$$

$$s.t.C_u = \sum_{C \in C_S, C \cap nb(u) \neq \phi} C$$

where $nb(u) = \{v : (u, v) \in E\} \cup \{u\}$. Equations (11)–(13) are satisfied respectively by the influence spread $f_c(S)$.

$$f_c(S) \geq 0 \tag{11}$$

$$f_c(S_1) \leq F_c(S_2) \tag{12}$$

$$f_c(S_1 \cup \{v\}) - f_c(S_1) \geq f_c(S_2 \cup \{v\}) - f_c(S_2) \tag{13}$$

where all $v \in V$ and $S_1 \subseteq S_2 \subseteq S$.

The steps of the proposed approach for identifying the core member is shown in Algorithm 1.

---

**Algorithm 1** Boundary-based Community Detection Approach **(BCDA)**

---

**Require:** Nodes, Candidate nodes, Network $G$, Community.
**Ensure:** Core nodes/members.
 1: Number of Nodes $\rightarrow n$.
 2: Candidate Node Set $\rightarrow S$.
 3: Core Nodes/Members $\rightarrow C$.
 4: **START**
 5: $C \rightarrow NULL$.
 6: Calculate $f_c(U|C_0)$ of each node $u$ in $S$. // Marginal Influence
 7: Select node $u$ with $f_c(U|C_0)_{max}$ into $C_1$.
 8: Remove $u$ from $S$.
 9: **for** $j$ *from* 1 *to* $n$ **do**
10:    Calculate Maximum = $f_c(u|S_j)$, Corenode = $u$.
11:    **for** each node $p \in S$ **do**
12:      **if** $f_c(p|C_{j-1}) > Maximum$ **then**
13:        calculate $f_c(p|C_j)$.
14:        **if** $(f_c(p|C_j)) > Maximum$ **then**
15:          update Maximum = $f_c(p|C_j)$.
16:          Corenode = $p$.
17:        **end if**
18:      **end if**
19:    **end for**
20:    Add Corenode into $C_j$.
21:    Remove Corenode from $S$.
22: **end for**
23: Set of Corenodes = $C_j$.

---

## 5. Results and Discussion

The work is implemented using high-end workstations with a configuration of 16 Gb RAM and 32 parallel processors. The communities are made based on the number of retweets and the number of followers. The extracted dataset consists of 16 attributes, Screen Name, user-created, tweet created, favorite count retweeted by, retweet count, user ID, tweet ID, text, language, following, followers, hashtags, in reply to status id, in reply to user id, and Class attribute. The dataset was filtered based on the user name and the tweet. The network structure is detected based on the retweets. An edge is drawn if a user retweets the already detected propagandistic tweet. Another criterion for creating a link is based on the number of followers. After performing experimentation, the results showed that there is more than one influential/core node in a propagandistic community. Figure 6 shows the community structure with its core members. Therefore, we conclude that many propagandistic users are responsible for sharing propaganda with the masses on Social Networks. Another conclusion that can be drawn from the results is that compromised accounts may also be used for spreading propaganda.

The users who share this information have the least followers, but their posts are retweeted through compromised accounts. It was also found that most of the propagandistic information is shared during trending events around the globe, for example, at times of the COVID-19 pandemic.

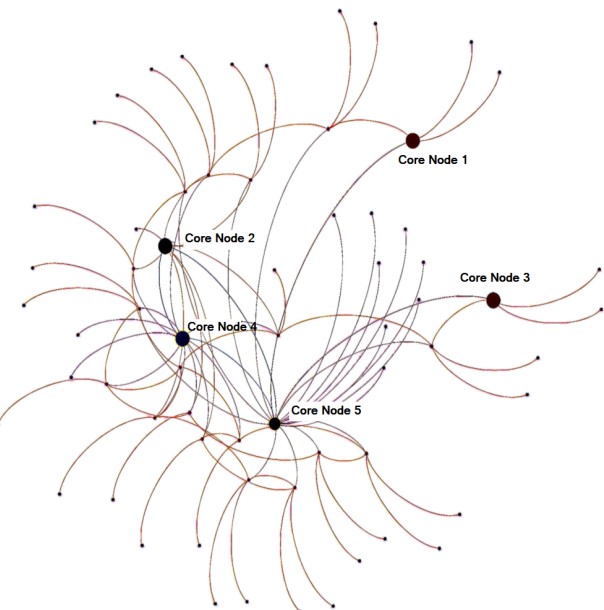

**Figure 6.** Community (C6) structure with their Core nodes.

*Comparative Study*

The previous researcher's work is compared with the proposed approach to validate this work. Table 1 shows the statistical features of the datasets that are used in experimentation. The following features are used to distinguish the datasets, Number of Nodes, Number of Edges, Average Degree ($D_{avg}$), Modularity, and Number of Communities formed. It was found that our approach outperformed other related works. Figure 7 shows the comparative analysis of various algorithms that are used for finding influence nodes. The algorithms which were used for comparison were:

- Greedy: Greedy algorithm is used to solve the problem of influence maximization. The greedy algorithm iteratively selects the nodes with the greatest marginal influence, due to which it has a very high-performance [49].
- ICRIM:To reduce the temporal complexity of Greedy, Improved Community-based Robust Influence Maximization (ICRIM) separates the network into multiple independent communities and then searches for core nodes within each community [50].
- CBIMA: Community-Based Influence Maximization Approach (CBIMA) identifies the influential nodes using community structure and influence distribution difference [51].
- Degree Discount: It is a heuristic algorithm based on the network structure [52].

The running time of each algorithm was calculated on these datasets; the result showed that the proposed approach outperforms other algorithms based on the running time. While running on the Citeseer dataset, the proposed approach took $10^2$ seconds to complete, while Greedy, ICRIM, and CBIMA took $10^4$, $10^3$, $10^{2.5}$ seconds, respectively, as depicted in Figure 7. The proposed approach (BCDA) achieved the best performance among all other algorithms. The Degree discount algorithm showed the least performance among all, and it can be concluded that heuristic algorithms only select core nodes based on network topologies' centrality index.

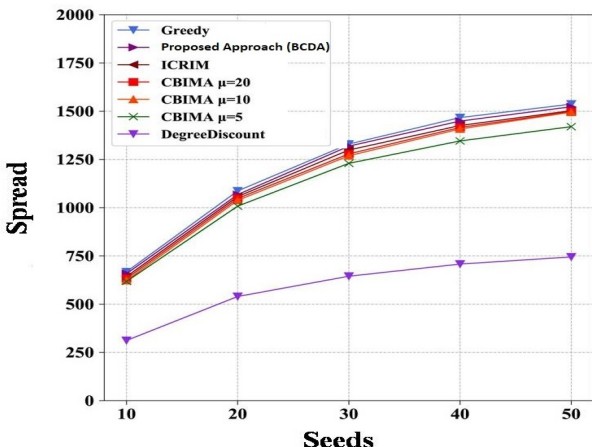

**Figure 7.** Comparison with existing approaches.

**Table 1.** Datasets used with their modularity score and number of communities found.

| Dataset | Number of Nodes | Number of Edges | $D_{avg}$ | Modularity Q | Communities C |
|---|---|---|---|---|---|
| Cora [53] | 2708 | 5429 | 4.02 | 0.6424 | 7 |
| Citeseer [54] | 3312 | 4732 | 2.86 | 0.4524 | 6 |
| Propt | 6500 | 13,000 | 3.7 | 0.490 | 6 |

## 6. Conclusions

Community detection on social networks has gained much interest from researchers around the globe. In this paper, we proposed an algorithm that detects propagandistic communities and the most influential node in that community. The algorithm consists of two phases first detecting the community using k-means clustering. In the second phase, identifying influential nodes, the nodes are selected from the interior and the propagandistic communities' boundary. The results showed that various communities share propaganda. On our dataset, we detected six propagandistic communities. It was also found that many influential nodes exist in the propagandistic communities. It can be concluded that various nodes are responsible for sharing propaganda on social networks. Due to the semantic nature of the text, propaganda identification is still in its infancy, and with an increase in the dataset, the complexity of the network increases. In the future, other algorithms can be used for detecting communities on social networks. Various existing approaches can be merged for detecting the community structure on social networks. Countering is another future direction in which work needs to be conducted.

**Author Contributions:** Conceptualization, A.M.U.D.K., M.A.W. and S.T.R.; Methodology, A.M.U.D.K. and M.A.W.; Validation, M.A.W.; Formal analysis, A.M.U.D.K.; Investigation, M.A.W., S.T.R. and Q.R.K.; Data curation, A.M.U.D.K. and M.A.W.; Writing—original draft, A.M.U.D.K. and M.A.W.; Writing—review & editing, A.M.U.D.K., M.A.W., S.T.R. and Q.R.K.; Visualization, A.M.U.D.K., M.A.W., S.T.R. and Q.R.K.; Supervision, M.A.W., S.T.R. and Q.R.K.; Project administration, Q.R.K.; Funding acquisition, M.A.W. All authors have read and agreed to the published version of the manuscript.

**Funding:** This research received no external funding.

**Institutional Review Board Statement:** Not applicable.

**Informed Consent Statement:** Not applicable.

**Data Availability Statement:** Not applicable.

**Acknowledgments:** The authors would like to acknowledge Prince Sultan University (PSU) and EIAS Data Science and Blockchain Laboratory for their valuable support. Also, the authors would like to thank Prince Sultan University for funding the Article Process Charges (APC) of this publication.

**Conflicts of Interest:** The authors declare no conflict of interest.

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
