# Peer review of "Hybrid Approach for Detecting Propagandistic Community and Core Node on Social Networks"

_sustainability, doi:10.3390/su15021249_

Round 1

Reviewer 1 Report

After careful consideration, I fell that the manuscript entitled "Hybrid approach for detecting propagandistic community and core node on social networks" has merit, but is not suitable for publication as it currently stands. Therefore, my decision is "Major Revision."

Here are my comments:

- Abstract: The authors say that "About six propagandistic communities are detected. The core members of the propagandistic community are a combination of a few nodes."However, there is no information about the data that generated these results.

- Section 3.2: The TK notation of the transition matrix is confusing. Sometimes the "k" is subscripted and sometimes not. Also, it may look like “T x k”. One suggestion is to use the notation T(k), for example

- standardize the subscripts "i,j". Sometimes it is "ij" and other times "i,j". The correct is the second way

- equation (2): Equation 2 has some problems:
    * What is the index (and interval of the summation?), i, j or k?
    * The notation T_{ki,j} is not correct. It should be T(k)_{I,j}
    * Something is wrong with "2dlnl = i".
    * Describe what "S(i,j)" is

- lines 200 and 226: standardize "O(m2n)" and "O(2mn)".

- Figures 1-7: Include the source of the figure (if you are the author, you can describe it as "source: Own Elaboration").

- Equation 3: Why was the number of edges adopted as "|E|" instead of "m" as adopted in the previous sections? I believe it is more interesting to standardize the notation of the whole manuscript.

- equation 5: describe what "N" means

- equation 6: what is node q_{i,l}? (since q is the summation index in Eq.(6))

- Figure 6. Figure 6 should be in the Results and Discussion section.

- line 12 - Algorithm 1: missing ")"

- Section 5.1. Include references to the algorithms Greed, ICRIM, CBIMA and Degree Discount.

- Table 1:
     * Table 1 is not cited and commented in the text.
     * What is "D_avg?"
     * In adition, there is no information and references of the datasets used.

Author Response

Response to Reviewer comments

*************************** Reviewer#1 *************************************

Concern#1: The authors say that "About six propagandistic communities are detected. The core members of the propagandistic community are a combination of a few nodes."However, there is no information about the data that generated these results.

Author’s Response:

Thank you so much for pointing out this deficiency. The Novel Dataset used in the proposed study has been extracted from Twitter using the python script (holding Tweepy package) and Twitter API.  

Author’s Action: 

As per the reviewer comments the required information about the dataset (which have been collected and used for the community detection)  has been added to several sections including abstract, data collection, comparative analysis, etc. in the manuscript. 

………………………………………………………………………………………………………

Concern#2: The TK notation of the transition matrix is confusing. Sometimes the "k" is subscripted and sometimes not. Also, it may look like “T x k”. One suggestion is to use the notation T(k), for example - standardize the subscripts "i,j". Sometimes it is "ij" and other times "i,j". The correct is the second way

Authors Response:

We appreciate the reviewer’s suggestion, the notations (TK ) are modified as suggested by the reviewer.

Author’s Action:

As suggested by the reviewer, the required information has been added to section 3.2 in the revised manuscript.

 ………………………………………………………………………………………………………

Concern#3: Equation 2 has some problems: * What is the index (and interval of the summation?), i, j or k? * The notation T_{ki,j} is not correct. It should be T(k)_{I,j} * Something is wrong with "2dlnl = i". * Describe what "S(i,j)" is

Authors Response:

Thank you for the valuable comments. The interval for summation is l running from i to n. The standard notation T(k)_{i,j) is used in the revised manuscript. 2dlnl is the typo error and is rectified in the revised version.The Euclidean distance between row vectors corresponding to nodes i and j in a matrix is used to compute the similarity between i and j for k walk length and is represented by S(i,j).

Author’s Action:

As per the reviewers comment we have updated equation 2 and  and provided a brief overview of S(i,j) in the revised manuscript. All the changes are used with track-changes on and  are highlighted in red color.

 ………………………………………………………………………………………………………

 Concern#4: - lines 200 and 226: standardize "O(m2n)" and "O(2mn)"

Author response:

Thank you very much for your valuable comment to improve the quality of the paper. We standardize O(m2n) and O(2mn) into O(2en) where e represents the edges and n represents the nodes.

 Author Action:

 As suggested by the reviewer we have modified the manuscript by standardizing O(m2n) and O(2mn) into O(2en).

 ………………………………………………………………………………………………………

Concern#5: - Figures 1-7: Include the source of the figure (if you are the author, you can describe it as "source: Own Elaboration").

Author response:

Thank you very much for the necessary comments. All the figures in the manuscript are our own elaborations and are generated from the collected while performing experiments.

Author Action:

The figures are own Elaboration. Each figure has been described in the manuscript. 

………………………………………………………………………………………………………

Concern#6: - Equation 3: Why was the number of edges adopted as "|E|" instead of "m" as adopted in the previous sections? I believe it is more interesting to standardize the notation of the whole manuscript.

- equation 5: describe what "Ni" means

- equation 6: what is node q_{i,l}? (since q is the summation index in Eq.(6))

- Figure 6. Figure 6 should be in the Results and Discussion section.

Author response:

Thank you very much for these valuable comments which improve the quality of the Manuscript. We have standardized the notation of edges as ‘e’ in the whole manuscript instead of m.

In equation 5,  “Ni” is the adjacent node set of i. 

In equation 6: the q  is the common neighbor node of i and j.  

As per the suggestion by the reviewer  the Figure 6 is moved to the section Results and Discussion section.

Author Action:

All the updates suggested by the reviewer are marked with track changes and are highlighted in red color in the revised manuscript. 

………………………………………………………………………………………………………

Concern#7: - line 12 - Algorithm 1: missing ")"

Author response:

Thank you so much for pointing out this typo error. The missing “)” is added in the algorithm 1, line 12. 

Author Action:

The missing “)” is added in the revised manuscript and is highlighted in red color.

………………………………………………………………………………………………………

Concern#8:- Section 5.1. Include references to the algorithms Greed, ICRIM, CBIMA and Degree Discount.

Author response:

Thank you so much for this comment which improves the quality of the paper 

Author Action:

As suggested by the reviewer the references are added to the algorithms : Greedy, ICRIM, CBIM and Degree Discount.

………………………………………………………………………………………………………

 Concern#9: - * Table 1 is not cited and commented in the text.

* What is "D_avg?"

* In adition, there is no information and references of the datasets used.

Author response:

Thank you so much for pointing out this deficiency. Table 1 is cited in the revised manuscript. D-Avg is the Average degree of the nodes. And datasets are also cited in the revised manuscript.

Author Action:

As suggested by the reviewer the table and datasets are cited in the updated manuscript. D_avg has also been discussed and is highlighted in red color in the revised manuscript.

Thank you very much for your comments on the language of the paper. We have now gone through the whole manuscript again to correct the possible typos and in correct sentences. Furthermore, the Grammarly-Professional tool has been used to correct the language, incorrect sentences, and misspelled words.

All the changes to the manuscript have been marked up using the “Track Changes” function in the LaTeX file. Some Changes are highlighted in red color in the revised.pdf file.

The authors would like to thank the reviewer for their valuable time to provide feedback and suggestions to improve the quality of our paper.

Reviewer 2 Report

The paper presents a hybrid approach for detecting propagandistic communities and

ore nodes on social networks.

Section 1 is the introduction. It is quite general and explained concepts which are well know, such as Twitter. It will be interesting to include a final paragraph in introduction stating the structure of the paper.

The paper presents in sections 2 and 3 the related work. Although many works are addressed and explained, those works are not compared with the proposal, so it is not clear how the announced contributions of the paper are obtained. Besides, the final conclusions of section 2 are vague and very general and do not contribute to show the relevance of the paper. 

In section 4, the model proposed in explained and compared with other systems. There are some important characteristics of the systems which are not explained with sufficient detail. For instance (line 279) “The extracted dataset consists of 16 attributes, Screen Name, date, retweeted by, retweet count, etc”. It seems relevant to state all the 16 attributes used.

There are some acronyms which are not explained nor referenced such as:  CBIMA (Line 304). The rest of algorithms introduced in this paragraph are not referenced (and they, as far as I know, are not introduced previously)

In general terms, the paper clearly presents the proposal. Nevertheless, Conclusion sections are  short and may be improved by adding more detail in what are the contributions of the paper and the proposal. Moreover, it is important to introduce what are the drawbacks of the system and the future work must be described. 

Finally, a general revision of the English language  must be done. Although, the paper is understandable, there are some grammatical errors that can be improved. 

Moreover there are sentences such as:

a) Line 48 “An Novel frTwitter * is proposed for detecting propagandistic communities and propagandists **” which seems incorrect. Something is lacking, in * what is it? algorithm, system, solution?, whereas in **, is it a node, a net?.

b) Line 60. “Finding a faster version of 60

he algorithm, the utilized algorithm becomes intractable for larger systems.”  I do not understand what the authors mean.

c) Line 67. “The aim of this work in the future is to find a faster version of the algorithm since it becomes intractable for larger systems and the complexity can be 

educed.” I do not understand what the authors mean, it seems that the sentence does not introduce relevant information.

Author Response

********************** Reviewer#2 ********************************

Concern#1: Section 1 is the introduction. It is quite general and explained concepts which are well know, such as Twitter. It will be interesting to include a final paragraph in introduction stating the structure of the paper.

Author response:

We are highly thankful for your valuable and insightful comment to include a paragraph about the structure of the paper. We have included the structure paragraph in the introduction section of the manuscript.

Author action:

As per reviewer's suggestion, we have updated the manuscript by adding a new paragraph about the structure of the paper. The newly added content can be seen in section 1 (Introduction) and is highlighted in red color.

………………………………………………………………………………………………………

Concern#2: The paper presents in sections 2 and 3 the related work. Although many works are addressed and explained, those works are not compared with the proposal, so it is not clear how the announced contributions of the paper are obtained. Besides, the final conclusions of section 2 are vague and very general and do not contribute to show the relevance of the paper.

Author response:

We are highly thankful for your valuable and insightful comment. Section 3 discusses about the background knowledge and few traditional approach that are used for detecting community structure. For comparative analysis recent related work is used to compare the proposed approach with them. Recent work is being cited in Related work, Background knowledge and result section.

Author action:

We have updated the manuscript by adding more recent work related to community detection and more references are added in section 3. Also the conclusion drawn from section 2 are updated in the revised manuscript and are highlighted in red color.

………………………………………………………………………………………………………

Concern #3: In section 4, the model proposed in explained and compared with other systems. There are some important characteristics of the systems which are not explained with sufficient detail. For instance (line 279) “The extracted dataset consists of 16 attributes, Screen Name, date, retweeted by, retweet count, etc”. It seems relevant to state all the 16 attributes used.

Author’s Response:

Thank you very much for your comment which definitely improves the quality of the paper.  We have added all the attributes of the extracted dataset.

Author Action:

As per the suggestion of the reviewer, we have updated the manuscript by adding all 16 attributes: Screen Name, user-created, tweet created, favorite count retweeted by, retweet count, user ID, tweet ID, text, language, following, followers, hashtags, in reply to status id,            in reply to user id and Class attribute and the same is highlighted in red color.

………………………………………………………………………………………………………

Concern #4: There are some acronyms which are not explained nor referenced such as: CBIMA (Line 304). The rest of algorithms introduced in this paragraph are not referenced (and they, as far as I know, are not introduced previously)

Author’s Response:

Thank you very much for pointing out these deficiencies. We have tried our best to add all the acronyms.

Author Action:

As per the valuable suggestion of the reviewer, we have updated the manuscript by adding acronyms and references to all the abbreviations that are used  in the manuscript. Changes are highlighted in Red color.

 ……………………………………………………………………………………………………

Concern #5: In general terms, the paper clearly presents the proposal. Nevertheless, Conclusion sections are short and may be improved by adding more detail in what are the contributions of the paper and the proposal. Moreover, it is important to introduce what are the drawbacks of the system and the future work must be described.

Author’s Response:

Thank you very much for your positive feedback. The conclusion has been updated and drawbacks are also mentioned.

Author Action:

As suggested by the reviewer the conclusion section is modified by adding drawbacks and future scope in conclusion.

 ………………………………………………………………………………………………………

Concern #6: Finally, a general revision of the English language must be done. Although, the paper is understandable, there are some grammatical errors that can be improved.

Author’s Response:

Thank you very much for your valuable suggestion. We have used Grammarly tool for rectifying grammatical and spelling mistakes in the manuscript.

Author Action:

As suggested by the reviewer the manuscript is being updated by rectifying spelling and grammatical errors and the Grammarly-Professional tool has been used to correct the language, incorrect sentences, and misspelled words.

All the changes to the manuscript have been marked up using the “Track Changes” function in the LaTeX file. Some Changes are highlighted in red color in the revised.pdf file.

The authors would like to thank the reviewer for their valuable time to provide feedback and suggestions to improve the quality of our paper.

………………………………………………………………………………………………………

Reviewer 3 Report

Thanks to the author for this interesting work. Will like them to take a read on my views and comments with respect to this work.

Hybrid approach for detecting propagandistic community and core node on social networks

This work proposes method of detecting propagandistic messages on a social network by using an algorithm that can detect both the sender and the community on social network.

Abstract:

This line - " The information shared can be dubious, detecting such kind of dubious network structure is need of an hour."

Reads a bit confusing and requires rewriting.

In Twitter, there are three types of API: Search API, Streaming API, and REST API. Use of acronyms - API, REST API without explanation isn't right.

Related Works

"Humans have the nature of making communities in the real world and the same is reflected in social media" - is this in social media or on social media?

These lines are confusing to read:

"The researchers in [14] identified the inner circles of government political power holders underneath formal work relations and observed how the selected political groups form and change over time work Construction, Community Discovery, and Community Evolution

Tracking were the three key processes they used to accomplish this."

Are there reasons for capitalising these words as seen here: "A study [15] developed a novel method for detecting Stealthy accounts in online social networks. Node-level community identification, Features, classification, and Finding Stealthy

Sybils are some of the steps in their approach."

In 3.1 and 3.2, there were only one citation each and do these imply no other researcher had in the past applied the use of the algorithm? How do you substantiate that the said algorithms will perform optimally without any earlier reference to feedback on the performance? The formulae too had no citation or source, are they that of the authors?

Few recent literatures are in the references. The authors should show that the work they are doing have not been overtaken by events by providing recent advancements in this domain.

Author Response

*********************  Reviewer#3 *********************************************

Concern#1: This line - " The information shared can be dubious, detecting such kind of dubious network structure is need of an hour." Reads a bit confusing and requires rewriting.

Author response:

We are highly thankful for your valuable and insightful comment by the reviewer; we have rephrased the sentence wherever possible in the manuscript to make it more readable and understandable.  

Author action:

We have updated the manuscript by modifying the existing sentence in the revised manuscript in abstract section.

………………………………………………………………………………………………………

Concern #2: In Twitter, there are three types of API: Search API, Streaming API, and REST API. Use of acronyms - API, REST API without explanation isn't right.

Author’s Response:

Thank you very much for your comment.  The acronyms of REST, API are added.

Author Action:

As per the reviewers suggestion, we have updated the manuscript by adding acronyms Application Program Interface (API) and Representational State Transfer (REST) in the introduction section.

 ………………………………………………………………………………………………………….

Concern #3: "Humans have the nature of making communities in the real world and the same is reflected in social media" - is this in social media or on social media?

Author response:

Thank you very much for identifying the typos to make the quality of the paper better.

Author Action:

We have updated the manuscript by modifying content from “Humans have the nature of making communities in the real world and the same is reflected in social media” to Humans have the nature of making communities in the real world and the same is reflected on social media”

………………………………………………………………………………………………………

Concern#4: These lines are confusing to read: "The researchers in [14] identified the inner circles of government political power holders underneath formal work relations and observed how the selected political groups form and change over time work Construction, Community Discovery, and Community Evolution Tracking were the three key processes they used to accomplish this." Are there reasons for capitalising these words as seen here: "A study [15] developed a novel method for detecting Stealthy accounts in online social networks. Node-level community identification, Features, classification, and Finding Stealthy Sybils are some of the steps in their approach."

Author response:

Thank you for the comments. We have modified the lines and uneven capitalizing of words are removed. The content is now more readable and makes sense.

Author Action:

As suggested by the reviewer, we have updated the manuscript by modifying the sentences that were used in the related work section.

………………………………………………………………………………………………………

Concern#5: In 3.1 and 3.2, there were only one citation each and do these imply no other researcher had in the past applied the use of the algorithm? How do you substantiate that the said algorithms will perform optimally without any earlier reference to feedback on the performance? The formulae too had no citation or source, are they that of the authors? Few recent literatures are in the references. The authors should show that the work they are doing have not been overtaken by events by providing recent advancements in this domain.

Author response:

Thank you very much for pointing this out. We have tried our best to add more proposed-work related citations for the background knowledge section. The proposed approach is also compared with the other existing approaches in the literature.

Author Action:

As per the suggestions of the reviewer we have updated the manuscript by adding more citations to the background knowledge section. Few more related references have been added to the revised manuscript.  

Thank you very much for your comments on the language of the paper. We have now gone through the whole manuscript again to correct the possible typos and in correct sentences. Furthermore, the Grammarly-Professional tool has been used to correct the language, incorrect sentences, and misspelled words.

All the changes to the manuscript have been marked up using the “Track Changes” function in the LaTeX file. Some Changes are highlighted in red color in the revised.pdf file.

The authors would like to thank the reviewer for their valuable time to provide feedback and suggestions to improve the quality of our paper.

………………………………………………………………………………………………………

Round 2

Reviewer 1 Report

Some of the changes made to the manuscript were not highlighted in red color. Even so, it was possible to see that all the suggested changes were made.

The authors made the corrections (or argumentations) satisfactorily. Therefore, my decision is to accept the paper.

Author Response

Response to Reviewer comments

*************************** Reviewer#1 *************************************

Comments:

Some of the changes made to the manuscript were not highlighted in red color. Even so, it was possible to see that all the suggested changes were made.

The authors made the corrections (or argumentations) satisfactorily. Therefore, my decision is to accept the paper.

Authors: Thank you very much for your time and effort in reviewing our manuscript and helping us to enhance the quality of the overall paper.

All the authors are grateful for your insightful comments on our paper.

Reviewer 2 Report

Authors have addressed almost all the changes proposed.

There is still the necessity of clearly stating what are the contributions of the paper, because comparison with the related work is still missing.  Nevertheless, the paper has been improved.

Author Response

********************** Reviewer#2 ********************************

The authors have addressed almost all the changes proposed.

Authors: Thank you very much for your time and effort in reviewing our paper.

All the authors are grateful for your insightful comments to enhance the quality of the overall paper

Comment 1: There is still the necessity of clearly stating what are the contributions of the paper, because comparison with the related work is still missing.  Nevertheless, the paper has been improved.

Author response:

Thank you very much for your input on the contribution of the paper. We have now updated the introduction of the paper by including the main contributions of the paper. The updated portion is highlighted with a red color and also marked up using the “Track Changes” function in the LaTeX file.

Author action:

The manuscript is now updated with the main contributions of the paper in a more readable way. Furthermore, we have named the proposed approach the Boundary-based Community Detection Approach (BCDA) in order to make clear the novel contributions of our paper. The following content has been added to the manuscript

  • An AI-based framework is proposed for detecting propagandistic communities and propagandists.
  • Leader Ranker Algorithm is used for mining the candidate nodes within the interior of a community.
  • Constraint Coefficient is used for mining candidate nodes within the boundary of a
  • The Boundary-based Community Detection Approach (BCDA) has been proposed to identify the core nodes in a propagandistic community based on the interior and boundary nodes which over-performed the existing approaches such as Degree Discount, Greedy, etc. in terms of running time.
  • A novel dataset collected from the Twitter blogging site is generated out of this study. This dataset consists of 16 attributes considered favorable for propagandistic community detection research. The dataset will be available for the researcher of this

Also, the manuscript and Figure 7 (Comparison with existing Approaches) have been updated as per the newly added content to the manuscript.

……………………………………………………………………………………………………….

All the changes to the manuscript have been marked up using the “Track Changes” function in the LaTeX file. Some Changes are highlighted in red color in the revised.pdf file.

The authors would like to thank the reviewer for their valuable time to provide feedback and suggestions to improve the quality of our paper.

………………………………………………………………………………………………………
